# Differential Modulation of Brassinosteroid and Ethylene Signalling Systems by Native and Constitutively Active Forms of the *AtCPK1* Gene in Transgenic Tobacco Plants Under Heat Stress

**DOI:** 10.3390/plants14071032

**Published:** 2025-03-26

**Authors:** Olga A. Tikhonova, Valeria P. Grigorchuk, Evgenia V. Brodovskaya, Galina N. Veremeichik

**Affiliations:** Federal Scientific Centre of the East Asia Terrestrial Biodiversity, Far East Branch of the Russian Academy of Sciences, 690022 Vladivostok, Russia; olga.tikhonova.kam@mail.ru (O.A.T.); kera1313@mail.ru (V.P.G.); aliedora2013@mail.ru (E.V.B.)

**Keywords:** AP2 transcription factors, brassinosteroids, calcium-dependent protein kinases, ethylene, heat tolerance

## Abstract

Among other calcium decoders, Ca^2+^-dependent protein kinases (CDPK) stands out for its ability, depending on calcium levels, to activate key components of the defence system. However, calcium dependence prevents the effective use of CDPKs in comprehensive investigations of their functions. Previously, we showed that a modified constitutively active form of AtCPK1 improved heat tolerance in tobacco plants. At present, the role of calcium ions and their decoders in the regulation of heat tolerance is not fully understood. The response of plant cells to excessive temperature increases is regulated by complex interactions of hormonal signalling systems, among which the least studied is BR signalling. In the present work, we investigated the role of CDPK in the interactions of BR and ET signalling during heat stress. The use of a modified calcium-independent form of AtCPK1 in this work allowed us to answer a number of questions. We showed that dependence on heat-induced calcium ion currents determines the priority of the activation of ABA signalling. Thus, CPK-dependent activation of ABA signalling may not lead to an insufficient response from BR and ET signalling. Modified CPK1 activates BR signalling, which has a positive effect on the tolerance of transgenic plants to increased temperature. The obtained data shed light on heat-associated molecular processes and can draw attention to the possibility of using intradomain modifications of CDPK both for a comprehensive study of its functional features and as a bioengineering tool.

## 1. Introduction

The response of plant cells to stress often depends on unique spatiotemporal fluctuations in intracellular calcium ions. Between 100 and 200 nM is the starting concentration of free calcium ions ([Ca^2+^]_cyt_) in plant cells’ cytoplasm. Depending on the type and strength of the external stimulus, the ‘induced increase in [Ca^2+^]_cyt_’ provides a unique dynamic known as the cytosolic ‘Ca^2+^ signature’ [1,2]. Calcium-dependent enzymes, in turn, decode [Ca^2+^]_cyt_ signals [3]. Along with calcineurin B-like proteins (CBLs) and calmodulin (CaM), the three primary families of [Ca^2+^]_cyt_ sensors in plants are Ca^2+^-dependent protein kinases (also known as CDPKs or CPKs). Compared with those of CaM and CBL, the unique structure of CDPKs provides the ability to mediate substrate-specific activation of target enzymes through phosphorylation [4]. In response to external biotic or abiotic stimuli, CDPK can modify responses regulated by stress-related hormones, including jasmonic (JA), abscisic (ABA), and salicylic (SA) acids, and ethylene (ET) [5]. However, little is known about the involvement of CDPK in brassinosteroid (BR) signalling.

Since 1998, BRs have been considered class 6 plant hormones. BRs are involved in the regulation of plant growth, development, and stress tolerance. Modulation of endogenous BRs can increase crop yield and quality as well as improve crop stress tolerance [6]. BRs are structurally similar to animal steroids [7]. BRs interact with other hormones to regulate adaptation to abiotic stresses. The BR receptor (BRASSINOSTEROID INSENSITIVE 1, BRI1) recognises the BR signal through the extracellular domain. Activated BRI1 transmits BR signals to protein signalling kinases (BSKs), which leads to BIN2 dephosphorylation. BIN2 is restrained by KIB1 (KIK SUPPRESSED IN BZR1-1D), which prevents the association of BINs with BZR1/BES1 and facilitates its ubiquitination [8]. BZR1/BES1 is dephosphorylated by PP2A (PHOSPHATASE 2A) and enters the nucleus. In this way, the regulatory role of BRs, in regulating the processes of growth, development, and resistance to osmotic and temperature stress, is realised. However, while the molecular mechanism by which BRs are involved in the regulation of the response to osmotic stress and cold is known, the role of BRs in heat stress remains to be elucidated. BR may be inhibited under heat stress, leading to the accumulation and activation of BIN2, which leads to the activation of *ERF49* (DREB2D) expression [9]. In another work, the authors proposed that BR is induced by heat stress, leading to the activation of BZR1, which represses *ERF49* expression. *Arabidopsis* HSFA1s function as “master regulators” and are induced under heat stress to directly regulate the expression of downstream *HEAT STRESS RESPONSIVE* (*HSR*) genes, or indirectly, through activation of the AP2/ERF TF *DREB2A* [10]. The expression of *ERF95* or *ERF97* enhances basal thermotolerance in *Arabidopsis* [11].

Thus, the link between BR signalling and thermotolerance is mediated by transcription factors of the AP2 family. Many transcription factors (TFs) mediate stress-responsive gene expression, and are altered by stress signalling. Arabidopsis has more than 2500 TFs [12]. Plant-specific transcription factors of the APETALA2/ethylene-responsive element binding factor (AP2/ERF) family have been demonstrated to control plant tolerance by altering downstream gene expression in response to stress [13]. AP2, dehydration-responsive element-binding protein (DREB), ethylene-responsive element-binding factor (ERF), ABI3/VP1 (RAV), and the unclassified Soloist subfamily are the five main subfamilies of the AP2/ERF family of proteins. ET signalling and the heat stress response have been linked in recent research [14]. The basal thermotolerance of Arabidopsis is improved by the overexpression of *ERF95* or *ERF97* [11,14]. The ethylene-responsive element (ERE) or GCC box with the AGCCGCC core sequence, which is found in the promoter/regulatory regions of genes responsive to ET, pathogens, and wounding, is where ERFs specifically bind and are linked to plant hormone ET responses [15]. The dehydration-responsive elements (DREs) or C-repeat elements (CRTs) with an A/GCCGAC core sequence found in the promoters of ABA-, drought-, and cold-responsive genes are bound by dehydration-responsive element-binding proteins encoded by DREBs. On the other hand, Arabidopsis DREB2s, such as DREB2A and DREB2B, react to drought, salt, and high temperatures rather than cold stress.

Thus, in the least studied area in this direction, the interaction of CDPK and BR can be called the process of adaptation to heat stress. As shown in recent studies, the most promising possible connection is through ethylene. Interestingly, little is known about the role of CDPK in regulating the response to heat. Only plant resistance to osmotic stress increases when some CDPK genes are overexpressed [16,17]. The role of calcium-dependent protein kinases and Ca^2+^ in thermotolerance is yet unknown [18]. However, there is no evidence that plants overexpressing CDPK have a direct improvement on heat tolerance. There have been numerous conflicting reports about the role of Ca^2+^ in thermotolerance. The reason for this discrepancy is that CDPK depends on intracellular Ca^2+^ levels; autoinhibition makes CDPKs dormant at baseline [Ca^2+^]_cyt_ levels. [18]. One remarkable example of self-regulated, stress-inducible enzymes is the autoinhibition of CDPK, which provides an appropriate response to stress. Four domains make up the structure of CDPK. In addition to the Ser/Thr kinase domain, there is the N-terminal variable domain, which is in charge of the substrate’s localisation and specific recognition. The C-terminal calmodulin-like domain (CaM-LD) contains conserved calcium-binding motifs, and an autoinhibitory junction (J) domain, which stops kinase activity at the baseline level of Ca^2+^ [5]. Harper [19] and Huang [20] used site-directed mutagenesis to produce different mutant forms of the *AtCPK1* gene (http://www.uniprot.org/UniProt/Q06850, accessed on 1 February 1995, GenBank accession number, AT5G04870) of *Arabidopsis thaliana*. Two mutant types, however, are particularly intriguing for physiological research. Regardless of whether Ca^2+^ ions were present, the mutant version of KJM4 was inactive. Four amino acid residues in the C-terminal portion of the KJM4 autoinhibitory domain are substituted, resulting in complete CDPK inactivation; this precludes the CaMLD domain from binding to Ca^2+^ [20]. In Harper’s work, the mutant form of KJM23 had a six-amino acid substitution in the autoinhibitory domain’s core region, ensuring maximum enzyme activity without the need for Ca^2+^ [19]. Since the initial discovery that plant cells have CDPKs, AtCPK1 from Arabidopsis has been one of the most studied members [21]. AtCPK1 overexpression has been associated with increased tolerance to drought and salt, as well as increased synthesis of phytoalexins [16,17,22,23,24,25,26]. Furthermore, the production of reactive oxygen species (ROS) is regulated by AtCPK1 [17,26]. By phosphorylating ORESARA1/ANAC092 (ORE1), AtCPK1 can also control cell death [27]. *AtCPK1*-transformed cell cultures of *R. cordifolia* showed that secondary metabolism and abiotic stress tolerance were activated simultaneously [26].

Previously, we showed that, in contrast to the native form, overexpression of the modified calcium-independent form of the *AtCPK*1 gene provides increased tolerance to heat stress in transformed tobacco plants grown in soil. Moreover, we have shown that the same heat stress treatments had more destructive effects when exposed plants were grown in vitro [28]. In addition, we showed that calcium supplementation increased the salt tolerance of both transformed and untransformed WT tobacco plants [17]. However, how Ca^2+^ contributes to thermotolerance remains unclear [18]. In the present work, we investigated the combined effects of heat treatment and calcium supplementation on the growth of tobacco plants and the germination of seeds under in vitro conditions. In the present work, we investigated the role of CDPK in the interactions of BR and ET signalling during heat stress. The use of a modified calcium-independent form of AtCPK1 allowed us to answer a number of questions. Both the effects of prolonged intense heat stress on adult plants and on seed germination were investigated. To understand the accompanying processes, the expression of key genes involved in the biosynthesis and signalling of BRs and ET was studied. The acquired information clarified heat-related molecular mechanisms and may highlight the potential of intradomain CDPK alterations for a thorough examination of its functional characteristics as a bioengineering tool.

## 2. Results

### 2.1. Effects of Short-Term Moderate Heat Stress on In Vitro-Grown N. tabacum Plants Transformed with Native and Modified AtCPK1 Genes

In the present study, we performed a detailed analysis of the heat tolerance of AtCPK1-KJM23 tobacco plants growing in vitro. Control WT plants and plants transformed with native and modified forms of the *AtCPK1* gene were grown in vitro on MS media under control conditions (24 °C) for 30 days, followed by moderate heat exposure (40 °C) for 10 days (Figure 1). Heat treatment caused leaf yellowing, which is associated with stress-induced senescence. As shown in Figure 1, the degree of yellowing of the leaves was significantly lower for KJM23-OE plants than for WT, KJM4-OE, and CPK1-OE plants (Figure 1).

We examined the treated and untreated plants’ relative NCC/DNCC contents (Figure 2) in order to validate this conclusion. Prior research has employed an analytical HPLC-UV/Vis-MS (MS2) approach to determine which chemicals had an increase in abundance during leaf senescence. Accordingly, two colourless compounds were previously identified and described as nonfluorescent chlorophyll catabolites (NCCs) with distinctive UV/Vis absorbance profiles [28]. Variations in the regions of the examined peaks served as the basis for the quantification computations (Figure 2A). The NCC/DNCC content in the WT, KJM4-OE, and CPK1-OE strains rose more than 3-fold following heat stress, according to analysis, but the NCC/DNCC level in KJM23-OE remained unchanged. These findings suggest that while overexpressing the native form of the AtCPK1 gene had no effect, overexpressing the calcium-independent form of the gene reduced heat-induced senescence in both soil and in vitro settings.

### 2.2. Combined Effects of Intense Heat Stress and Calcium Supplementation on the Growth and Accumulation of Chlorophyll Catabolites of N. tabacum Plants

Untransformed WT and transgenic tobacco plants were grown in vitro for 20 days under control conditions (24 °C). To determine the effect of calcium supplementation, plants were grown on media supplemented with 0, 1.5, or 10 mM CaCl_2_. Twenty-day-old plants were exposed to moderate heat stress at 40 °C for 20 days. As shown in Figure 3A, Ca^2+^-free conditions prevented heat-induced damage to chlorophyll in heat-treated WT and KJM4-OE plants, while a negative effect of Ca^2+^ supplementation was observed.

We analysed the relative NCC/DNCC content (Figure 3B) in the treated and untreated plants.Plants grown on the standard MS medium contained 1.5 mM CaCl_2_ under unstressed conditions (24 °C) for 40 days were used as controls. Analysis revealed that after heat stress, the NCC/DNCC content in the WT, KJM4-OE, and CPK1-OE plants increased more than 3-fold, whereas the NCC/DNCC content in KJM23-OE did not change when the plants were grown on media supplemented with standard 1.5 mM CaCl_2_. A dose-dependent negative effect of Ca^2+^ supplementation on the relative NCC/DNCC content in the WT, KJM4-OE and CPK1-OE plants was observed. The relative NCC/DNCC content (Figure 3B) was two times lower in the CPK1-OE and KJM23-OE plants than in the WT and KJM4 plants in the absence of additional calcium. However, the beneficial effect of CPK1 is reduced when calcium levels are increased. The KJM23-transgenic plants presented an approximately 2-fold decrease in the NCC/DNCC content within the concentration range of 0–10 mM.

Next, we studied the effects of slight heat treatment and calcium supplementation on the germination (Figure 4) of WT and plants transformed with native and modified *AtCPK1* gene seeds. The seeds were sterilely germinated for 17 days in MS media supplemented with 0, 1.5, or 5 mM CaCl_2_ under control conditions (24 °C) or heat treatment (37 °C). A low concentration of CaCl_2_ and heat did not affect the percentage of germinated seeds; for all treated variants of tobacco plants, the percentage was approximately 80% (Figure 4A,B). These findings indicate the absence of a considerable inhibitory effect of slight heating on the process of germination. A decrease in germination was observed only for WT and KJM4 when the CaCl_2_ concentration was increased to 5 mM, whereas the germination of CPK1-OE and KJM23-OE seeds was not affected. However, the lengths of the seedlings was significantly shorter those that of the control group (Figure 4A,C). The reduction was approximately 4-fold under the combined effect of temperature and calcium, 3-fold under the effect of temperature alone, 2-fold under the effect of the highest calcium concentration, and 1.3-fold under a calcium concentration of 1.5 mM. This finding pointed to the stage-dependent action of recombinant AtCPK1. At the early stage of growth, we did not detect any differences between the native and modified constitutively active forms of the *AtCPK1* gene.

### 2.3. Thermoinduced Expression of BR Biosynthesis and Signalling Genes in AtCPK1-Expressing Tobacco Plants

Next, we studied the expression of the genes involved in BR biosynthesis (*NtCYPs* and *NtDSR*) and BR signalling (*NtBZR1* and *NtBR1*) in *N. tabacum* plants under control conditions and after heat shock treatment with or without CaCl_2_ (Figure 5). Analysis of gene expression was carried out on 40-day-old plants growing in vitro. In addition, the analysis was also carried out under intense short-term heat stress (42 °C) for 1 h. Furthermore, the impact of the combined effect of temperature and an increase in the calcium concentration in the medium was also investigated. An approximately 25-fold decrease in *CYP85A1* gene expression was observed in wild-type (WT) plants, as well as in KJM4-OE and CPK1-OE plants, in response to heat stress and the combined effects of temperature and calcium. In contrast, the results obtained for KJM23-OE plants revealed that there was no change in *CYP85A1* gene expression in response to short-term heat stress. However, the combination of calcium and heat led to a 2.6-fold decrease in *CYP85A1* gene expression. The effects of heat and calcium inhibited the expression of the *CYP90A1* gene 8-fold in the WT and KJM4 plants and 5-fold in the CPK1-OE and KJM23 plants. Additionally, a 5-fold decrease relative to the control conditions was observed when the plants were exposed to heat alone. The expression of the *NtCYP90B1* gene was similar in all the tested treated and untreated plants. The expression profile of the BR biosynthesis-related gene *NtDSR* was maintained at the level of the control under heat shock conditions for all the plants but decreased 2- to 3-fold when heat and calcium were combined (Figure 5).

The expression patterns of the BR signalling genes *NtBZR1* and *NtBRI1* in the leaves of wild-type (WT), inactive (KJM4), and native (CPK1-OE) *AtCPK1*-transformed tobacco plants were similar. Under control conditions, the expression of both genes and BR biosynthetic genes did not change in WT or transgenic tobacco plants. Heating, as well as the combination of heat and calcium supplementation, led to a more than 2-fold decrease in *NtBZR1* and *NtBRI1* gene expression in the WT, KJM4-OE, and CPK1-OE plants. In contrast, the expression levels of the BR signalling genes *NtBZR1* and *NtBRI1* in KJM23-OE plants remained unaffected by heat and calcium exposure (Figure 5).

### 2.4. Phylogenetic Analysis of the AP2 Family Members and the Design of Unique Primer Pairs

To determine the role of AtCPK1 in heat-induced crosstalk between BR and ET signalling, we performed qPCR analysis of the expression of transcription factors of the AP2 family involved in the heat response. This research was based on the described model for *A. thaliana* [29]. Among other genes of *N. tabacum*, we selected the closest homologues of the negative regulator AtEFR49/DREB2D and the positive regulators AtERF095 and DREB2A. In addition, we included in the analysis previously studied *NtERF* genes [30].

Before investigating the expression patterns, we performed an in-depth phylogenetic analysis of the selected genes. Amino acid sequences of the AP2 members of *A. thaliana*, *N. tabacum* and other plants used in this work were retrieved from the GenBank and UniProt databases and are listed in Appendix A. For the alignment of amino acid sequences, the ClustalW and ClustalX programmes were used. Multiple alignments of the amino acid sequences strongly divided the studied AP2s into four groups (ERF, DREB, RAV, and AP2) and highlighted them in green, pink, blue, and red blocks, respectively. The conserved single AP2 domains in ERFs and the DREB and RAV subfamilies are highlighted in green, pink, and blue blocks, respectively. The first conserved AP2 domain in the AP2 subfamily is highlighted with a red block. The AP2 family is divided into subfamilies (ERF, DREB, RAV, and AP2), and conserved domains are defined according to Choudhury [31]. As shown in Figure 6, NtERF1-5 and NtERF14 belong to the ERF subfamily, as does AtERF096. NtDREB2D and NtDREB2A belong to the DREB subfamily, as do AtERF49/DREB2D and AtDREB2A (Figure 6).

Using the ClustalW online tool, a phylogenetic tree was constructed on the basis of the alignment of amino acid sequences performed via ClustalX. As shown in Figure 7, NtERFs (NtERF1-5, NtERF14), homologues of *A. thaliana* DREB2A, DREB2D, ERF96, and homologues from other plants were divided into subfamilies (ERF, DREB, RAV, and AP2) and highlighted in green, pink, blue, and red blocks, respectively. The AP2 family members of *N. tabacum* and *A. thaliana* are designated with dots and triangles, respectively, and the colours correspond to the subfamily groups. As shown in Figure 7, NtERF1-5 and NtERF14 belong to the ERF subfamily, as well as AtERF096, and NtERF14 is closer to AtERF096 than the other analysed NtERFs are. NtDREB2A and NtDREB2D divided and clustered together with AtDREB2A and AtDREB2, respectively.

### 2.5. Thermoinduced Expression of ET Biosynthesis and Signalling Genes in AtCPK1-Expressing Tobacco Plants

We also investigated the expression of ethylene biosynthesis (*NtACS*) and signalling (*AP2* family) genes in *N. tabacum* plants after heat shock treatment with or without calcium supplementation (Figure 8). Analysis of gene expression was carried out on 40-day-old plants growing in vitro. In addition, the analysis was also carried out under intense short-term heat stress (42 °C) for 1 h. Furthermore, the impact of the combined effect of temperature and an increase in the calcium concentration in the medium was also investigated. The expression profile of the ET biosynthesis-related gene *NtACS* was 3.5-fold lower under heat shock conditions in wild-type (WT) plants and plants transformed with the native (CPK1-OE) and mutant inactive (KJM4-OE) forms of *AtCPK1*. The WT and KJM4 plants presented similar expression levels when exposed to both calcium and heat. When the effects of calcium and heat were combined on plants with native and calcium-independent forms of CPK, only a 1.5-fold decrease in expression for the native form was observed. In contrast, under these conditions, the expression in KJM23 plants remained unchanged relative to that in the control.

The present study examined the expression of the *ERF14* gene in KJM23 plants under standard conditions and revealed that it was expressed at 2.5-fold greater levels than the other forms and the wild type. Heat shock caused a 2-fold decrease in expression for the two tested forms and WT, and a 2.5-fold decrease for KJM23-OE. The additional effect of calcium essentially blocked the expression in the WT, KJM4-OE, and CPK1-OE plants and decreased the gene expression in the KJM23-OE plants 5-fold. The expression of the *ERF1* gene was blocked in the WT and KJM4-OE plants under heat shock conditions, the expression level in the CPK1-OE plants was unchanged compared with that in the control plants, and the transformation of KJM23 resulted in an approximately 5-fold increase. The highest expression of the *ERF2* gene, which was approximately 4-fold greater than that in the control, was detected in the KJM23-OE transgenic plants under heat shock on calcium-supplemented media. The expression of this gene was virtually blocked in the other forms studied. Exposure to heat alone caused a twofold increase in KJM23 plants and blocked the expression of the other forms. The expression of the *ERF3* gene was maintained at the highest level at the time of transformation in the KJM23 mutant under all the studied conditions. No changes in expression were observed in the other plants. The expression of the *ERF4* gene was consistently high at 2.5–5 times greater than that of the control gene when the KJM23 mutant was introduced under all the conditions studied. No expression changes were observed for WT or transgenic plants with native (CPK1-OE) or mutant inactive (KJM4-OE) AtCPK1. Temperature exposure resulted in a decrease in *ERF5* gene activity of approximately 11-fold for the WT, KJM4-OE, and CPK1-OE plants and 3.5-fold for the KJM23 plants. Compared with temperature alone, additional exposure to elevated calcium concentrations led to almost complete inhibition of gene expression in the wild type (WT) plants and two other forms (KJM4-OE and CPK1-OE), whereas the expression in KJM23 plants remained unchanged (Figure 8).

DREBs are related to stress resistance in a wide range of plant species. We analysed the expression of the *NtDREB* genes in *N. tabacum* plants after heat shock treatment at normal (1.5 mM) and high (10 mM) calcium concentrations. Analysis of gene expression was carried out on 40-day-old plants growing in vitro. We found that *DREB2A* gene expression was activated 5-fold in response to stress in WT and plants with added AtCPK1 KJM4 and KJM23 mutants, and approximately 3-fold in CPK1-OE plants. An increased calcium concentration in the medium decreased the activating effect of heat shock, but relative to that of the control, the calcium concentration increased by 2-fold for WT and KJM4 and 3-fold for KJM23. In addition, the expression of the *DREB2A* gene in CPK1-OE plants was nearly blocked under these conditions. When plants were injected with the native form of CPK1-OE, the expression of the *DREB2D* gene increased under both control conditions and under exposure to heat alone and heat together with calcium. The increases were approximately 4, 30, and 50 times, respectively. The other forms studied showed a weak response to changing conditions, and we observed only a 10-fold increase in expression in the WT and KJM4 plants in response to the combined effects of calcium and heat shock. The expression of *DREB2D* in KJM23 plants remained consistently low (Figure 9).

## 3. Discussion

At present, the role of calcium ions and their decoders in the regulation of heat tolerance is not fully understood. The responses of plant cells to excessive temperature increases are regulated by complex interactions of hormonal signalling systems, among which the least studied is BR signalling. In the present work, we investigated the role of CDPK in the interactions of BR and ET signalling during heat stress. The use of a modified calcium-independent form of AtCPK1 in this work allowed us to answer a number of questions. First, why does the overexpression of native forms of CDPK genes not provide resistance to heat to the same extent as to salt, drought, and cold? We showed that the dependence on heat-induced calcium ion currents determines the priority of the activation of ABA signalling. Thus, CPK-dependent activation of ABA signalling may not lead to an insufficient response from BR and ET signalling. Under conditions independent of heat-induced ion currents, modified CPK1 activates BR signalling, which has a positive effect on the tolerance of transgenic plants to increased temperature.

The impact of global warming makes crop production more challenging. The Intergovernmental Panel on Climate Change (IPCC) estimated that the average global temperature would rise by 0.3 °C over a ten-year period, and by 2025, it is expected to rise by approximately 1 °C [31]. Because heat stress upsets cellular homeostasis, which eventually leads to stunted development and even death, it poses a serious threat to crop productivity globally [32]. In the present work, we showed that a modified, constitutively active form of the *AtCPK1* gene confers tolerance to the negative effects of heat stress on transgenic tobacco plants, in contrast to the native form. However, in the early stages of development, during seed germination, both the native and modified constitutively active forms have similar effects on heat tolerance. This finding pointed to the stage-dependent action of recombinant AtCPK1. Recent research has shown that certain *CDPK* gene isoforms respond to heat shock [33,34,35,36,37,38]. Four of the nineteen isoforms of the *CDPK* genes of the wild Chinese grape *V. pseudoreticulata* are expressed more highly when exposed to short-term (2–48 h) (42 °C) heat stress [38]. Under severe short-term stress, wild soy presented significantly greater increases in *GmCDPK5* and *GmCDPK10* expression than cultivated soybeans did [37]. It has also been demonstrated that short-term heat treatment activates defensive mechanisms in plants overexpressing CDPK [39,40]. Under heat stress, *AtCPK28* should directly phosphorylate the stress marker enzyme APX2, leading to increased heat resistance [41]. Furthermore, the *cdpk* mutant is relatively insensitive to heat [42]. However, there is no evidence that plants with higher levels of CDPK overexpression are more heat tolerant [18]. Thus, the dependence of CDPK on calcium prevents its positive participation in increasing heat tolerance.

We have previously shown that similar heat stress has a more severe effect on plants grown in vitro than on those grown in soil [28]. Therefore, in the present work, we investigated the effects of heat stress in vitro, which allowed us to conduct valid studies of the effects of the Ca^2+^ concentration in nutrient media on heat tolerance. We showed that even a small concentration of calcium aggravated the negative impact of heat stress. The least damaging effect was shown in the absence of CaCl_2_ in the medium. During seed germination, an increase in the CaCl_2_ concentration also had a negative effect on the WT, while the transgenic seeds were resistant. The role of Ca^2+^ in thermotolerance has been the subject of numerous conflicting studies. The inability of CDPKs to provide tolerance to prolonged heat stress in plants overexpressing CDPK may be explained by the unknown involvement of Ca^2+^ in the cell’s reaction to heat stress. It has been demonstrated that heat-induced Ca^2+^ influx has a deleterious effect [43]. Furthermore, Ca^2+^ may control how programmed cell death (PCD) develops. Therefore, Ca^2+^ clearly has two functions in how plants react to high temperatures. On the one hand, it may save the plant from dying by activating the expression of heat shock proteins (HSPs); on the other hand, it may cause plant death. Feedback loops are thought to prevent the potential beneficial effects of CDPK. Currents or phosphatase activation can impede phosphorylation-regulated Ca^2+^ channels or Ca^2+^-induced phosphorylation-dependent processes. The SAR and ABA signalling systems may be antagonistic during this process [44]. In the present study, we focused on the least studied heat stress signalling system, BR signalling, and interconnected ET signalling.

We showed that heat stress leads to a decrease in the expression of BR biosynthetic genes. However, overexpression of the modified form of AtCPK1-KJM23 significantly inhibited this effect (Figure 10). This, in turn, ensures the stabilisation of the expression of genes encoding key BR signalling mediators, *BZR* and *BRI*, in KJM32-OE plants. Moreover, stabilisation of the expression of key players in BR signalling under the combined influence of heat and Ca^2+^ also increases the negative effect. However, the role of BRs in thermotolerance cannot be considered in isolation from ET signalling. We also analysed the expression of genes encoding transcription factors of the AP2 family, including ERFs and DREBs. During the early stages of heat exposure, the intracellular calcium concentration increases [45], which triggers cascades regulated by CDPK. BR may be inhibited under heat stress, leading to the accumulation and activation of BIN2, which leads to the activation of *ERF49* (DREB2D) expression [9]. In another work, the authors proposed that BR is induced by heat stress, leading to the activation of BZR1, which represses *ERF49* (DREB2D) expression. *Arabidopsis* HSFA1s function as “master regulators” and are induced under heat stress to directly regulate the expression of downstream *HEAT STRESS RESPONSIVE* (*HSR*) genes, or indirectly regulate it by activating the AP2/ERF TF *DREB2A* [15]. The expression of *ERF95* or *ERF97* enhances basal thermotolerance in *Arabidopsis* [11]. This model was described in detail by [15]. As we previously reported, overexpression of the native form of the AtCPK1 gene led to a heat-induced increase in the expression of genes involved in ABA biosynthesis, in contrast to the calcium-independent form KJM23.

We demonstrated that tobacco tolerance or sensitivity to heat stress is unaffected by *AtCPK1* overexpression. Furthermore, tolerance to extended heat exposure is markedly increased by the modified AtCPK1 form, which is independent of changes in intracellular Ca^2+^ levels. Nevertheless, nothing is known about how intracellular Ca^2+^ regulates heat tolerance. On the basis of the information gathered, inactivating the CDPK autoinhibitory domain appears to be a viable method for increasing a plant’s resistance to temperature stress.

## 4. Materials and Methods

### 4.1. Plant Materials

In the present work, previously obtained clonally cultivated plantlets of *Nicotiana tabacum* L. (cv Xanthi) plants were used [17]. Non-transformed tobacco plants were used as the control, and designated WT. Plants transformed with the native form of the *AtCPK1* gene (construction pPZP-RCS2-nptII/Ak) were designated CPK1-OE. Plants transformed with the nonactive form of the *AtCPK1* gene (construction pART27/AtCPK1-KJM4) were used as additional controls and designated KJM4-OE. Plants transformed with the active form of the *AtCPK1* gene (construction pART27/AtCPK1-KJM23) were designated KJM23-OE. The genetic construction [19,20,23] and transgenic plants [17] were described previously. As shown previously, the expression level of the *AtCDPK1* gene in the transgenic tobacco plants was the same [17]. Seeds of the third generation (T3) of transgenic *N. tabacum* plants were obtained previously [28]. The WT and *AtCDPK1*-transgenic plants were grown in vitro via MS agar media [46] under the following conditions: illumination in the daytime, 3000–5000 lux of luminescent lamps (approximately 150 µmol m^−2^ s^−1^), 16/8 h; temperature, 24/22 °C; and humidity, 70%.

### 4.2. Experimental Design

In the present work, heat-induced senescence was investigated in control and transgenic tobacco plants grown in vitro under the same conditions. Seeds of the control nontransgenic WT tobacco plants and seeds of the third generation of the two independent clones of each variant of the transgenic plants (CPK1-OE, KJM4-OE, and KJM23-OE) were used. Seeds were sterilised in ethanol, washed twice in sterile water and germinated in 200 mL Erlenmeyer flasks containing 50 mL of MS media. To investigate the effect of calcium supplementation, the medium was supplemented with 0, 0.5, 1.5 (standard concentration), or 10 mM CaCl_2_. Two types of experiments were performed: (i) seeds were germinated and grown for 20 days under control conditions (24 °C), followed by moderate heat exposure (40 °C for 20 days), and (ii) 30 days under control conditions (24 °C), followed by intense heat exposure (42 °C for 10 days). The corresponding control conditions were 40 days under the control temperature (24 °C). After the stress treatments, the experimental plants were photographed and harvested for the analysis of heat-induced phyllobilin accumulation. Both types of heat stress experiments were repeated three times. For analysis of germination under heat stress, seeds were germinated for 17 days under control conditions (24 °C) or slight heat exposure (36 °C) via media supplemented with 0, 1.5 (standard concentration), or 5 mM CaCl_2_.

*Molecular analysis*. For determination of gene expression, forty-day-old clonally cultivated control (WT) and transformed tobacco plants were transferred to 42 °C for 1 h with or without supplementation with 10 mM CaCl_2_. RNA was extracted immediately. In this work, we used three independent lines of control plants and each type of transgenic plant.

### 4.3. HPLC-DAD-ESI-MS(/MS) Conditions

#### 4.3.1. Chemicals

LC-MS-grade methanol and acetonitrile were acquired from Merck (Darmstadt, Germany). MS-grade formic acid was acquired from Sigma Aldrich (Steinheim, Germany). Deionized water was obtained via a Milli-Q Simplicity water purification system (Millipore, Molsheim, France).

#### 4.3.2. HPLC-UV Determination of the Breakdown Products of Chlorophyll

Phyllobilin extraction and quantification were performed as we reported previously [28]. Reversed-phase high-performance liquid chromatography with ultraviolet absorbance detection (RP-HPLC-UV) was performed for chlorophyll catabolite determination. A 1260 Infinity analytical HPLC instrument (Agilent Technologies, Santa Clara, CA, USA) was used for analysis. An analytical Zorbax C18 column (150 mm, 2.1 mm i.d., 3.5 μm; Agilent Technologies, Santa Clara, CA, USA) was used for chromatographic separation. UV spectra of the studied compounds were recorded with a DAD in the range of 200–500 nm, and chromatograms were extracted at a wavelength of 312 nm. Two colourless nonfluorescent chlorophyll catabolites (*Nr*-NCC-1 and *Nr*-NCC-2) were quantified on the basis of differences in the areas of the analysed peaks.

### 4.4. Phylogenetic Analysis

Alignment and phylogenetic reconstructions were performed via the function “build” of ETE3 3.1.3 [47] as implemented in the GenomeNet (https://www.genome.jp/tools/ete/, accessed on 3 April 2015). We provided multiple sequence alignment via ClustalX2 [48]. Multiple sequence alignment was performed via Clustal X. The tree was constructed via FastTree v2.1.8 with default parameters [49]. The values at the nodes represent bootstrap support.

### 4.5. Analysis of Gene Expression

Total RNA extraction was performed as described previously [18]. Before use, RNA samples were treated with DNAse (Biolabmix, Novosibisk, Russia) according to the manufacturer’s instructions. First-strand cDNA synthesis was performed as described previously [18]. The oligonucleotide pairs used as housekeeping genes (Actin and EF1α) and for the analysis of BR and ET biosynthesis and signalling gene expression are listed in Appendix A. Real-time PCR was carried out as described previously [38] under the following conditions: in a 96-well reaction plate, a 10 μL final volume comprising 1 μL of the diluted cDNA sample and 300 nM of each primer was used; the reaction conditions were as follows: 3 min at 96 °C, 35 cycles of 15 s at 96 °C, 15 s at 60 °C, and 30 s at 72 °C. Three technical replicates were evaluated for every biological replicate, and two biological replicates were used for the analysis. qPCR analysis was performed with both the RNA-RT and no-template controls to confirm that there was no contamination. Melting curve analysis at the end of each run was performed to establish the lack of primer-dimer artefacts or nonspecific products. The software CFX Manager (version 3.1) was used to analyse the data.

### 4.6. Statistical Analysis

All values are represented via Statistica 10.0 (StatSoft Inc., Tulsa, OK, USA) as the mean ± SE. A significance threshold of *p* < 0.05 was applied to the differences. ANOVA and multiple comparisons were used to compare data across different groups, and Student’s *t* test was used to analyse data between two independent categories. For the intergroup comparisons, Fisher’s protected least significant difference (PLSD) post hoc test was used. ANOVA tables are included in the Appendix A. Microsoft Excel software was used to develop the graphs.

## Figures and Tables

**Figure 1 plants-14-01032-f001:**
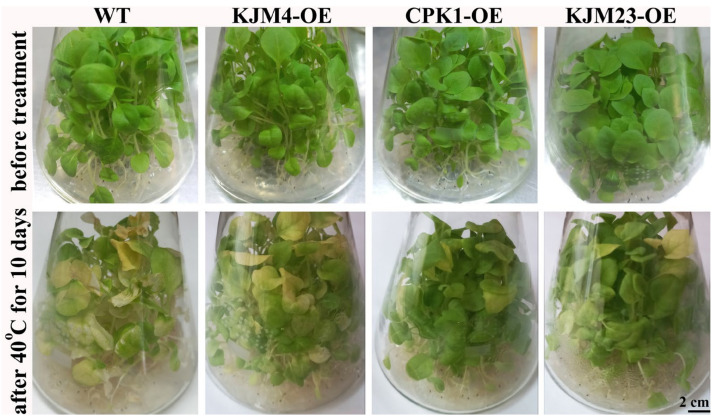
Short-term moderate heat stress’s effects on the morphology of *N. tabacum* plants cultivated in vitro. Bottom: plants grown under control conditions (24 °C, 30 days), followed by moderate short-term heat exposure (40 °C, 10 days); top: 40-day-old plants cultivated in vitro under control conditions (24 °C, 40 days). The *AtCPK1* gene in tobacco plants was cultivated in vitro in standard medium in the following forms: constitutively active (KJM23-OE), mutant inactive (KJM4-OE), and control (WT) and transformed native (CPK1-OE).

**Figure 2 plants-14-01032-f002:**
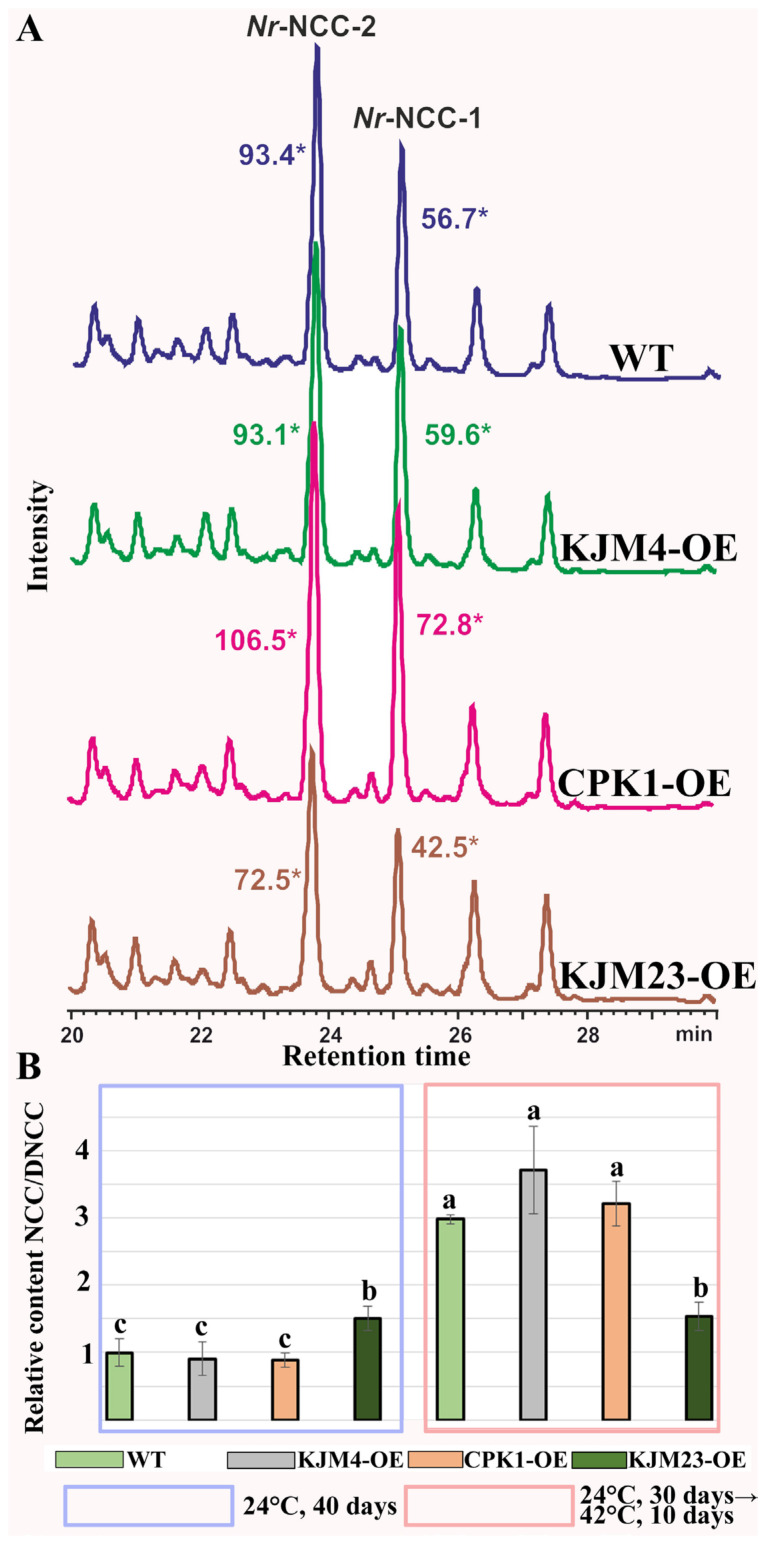
Heat-induced accumulation of NCC/DNCC chlorophyll catabolites in *N. tabacum* plants HPLC separation (**A**) and UV determination of chlorophyll catabolites from *N. tabacum* plant extracts. Chromatograms were recorded at 312 nm and are presented with an overlay and on the same scale. The coloured asterisks (*) indicate the peak areas of the studied compounds. NCC/DNCC (**B**) relative content in extracts of 40-day-old tobacco plants grown in vitro under unstressed conditions (24 °C, blue frame) and after heat treatment at 40 °C for 10 days (pink frame): native (CPK1-OE), mutant inactive (KJM4-OE), and constitutively active (KJM23-OE) forms of the *AtCPK1* gene-transformed plants and control plants (WT). Three separate experiments provided the results, which are shown as mean ± standard error of the mean. Statistically significant differences are indicated by the various letters above the error bars (*p* < 0.05, Fisher’s LSD).

**Figure 3 plants-14-01032-f003:**
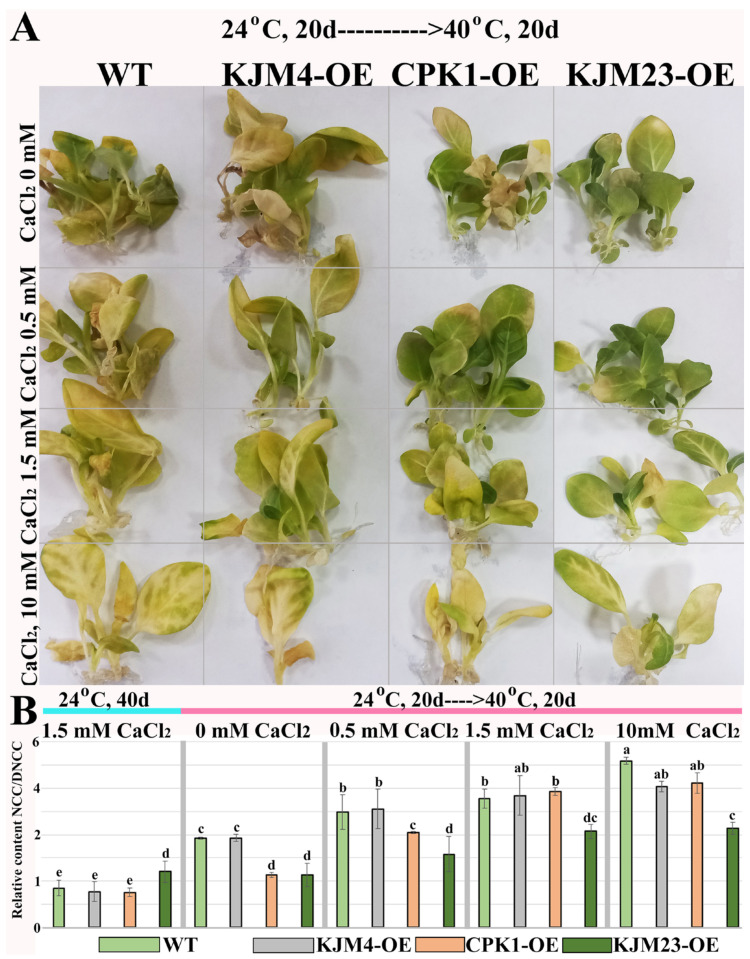
Combined effects of intense heat stress and calcium supplementation on the growth and accumulation of NCC/DNCC chlorophyll catabolites in *N. tabacum* plants. (**A**) 40-day-old plants grown in vitro under control conditions for 20 days followed by moderate heat exposure (40 °C, 20 days). Relative content of NCC/DNCC (**B**) in extracts of 40-day-old tobacco plants grown in vitro under unstressed conditions (24 °C, blue line) with standard calcium supplementation and after heat treatment at 40 °C for 20 days (pink line): the control plants (WT) and plants transformed with the native (CPK1-OE), mutant inactive (KJM4-OE), and constitutively active (KJM23-OE) forms of the *AtCPK1* gene. For heat treatments, the plants were grown in MS/2 media supplemented with 0, 1.5, or 10 mM CaCl_2_. Three separate experiments provided the results, which are shown as mean ± standard error of the mean. Statistically significant differences are indicated by the various letters above the error bars (*p* < 0.05, Fisher’s LSD).

**Figure 4 plants-14-01032-f004:**
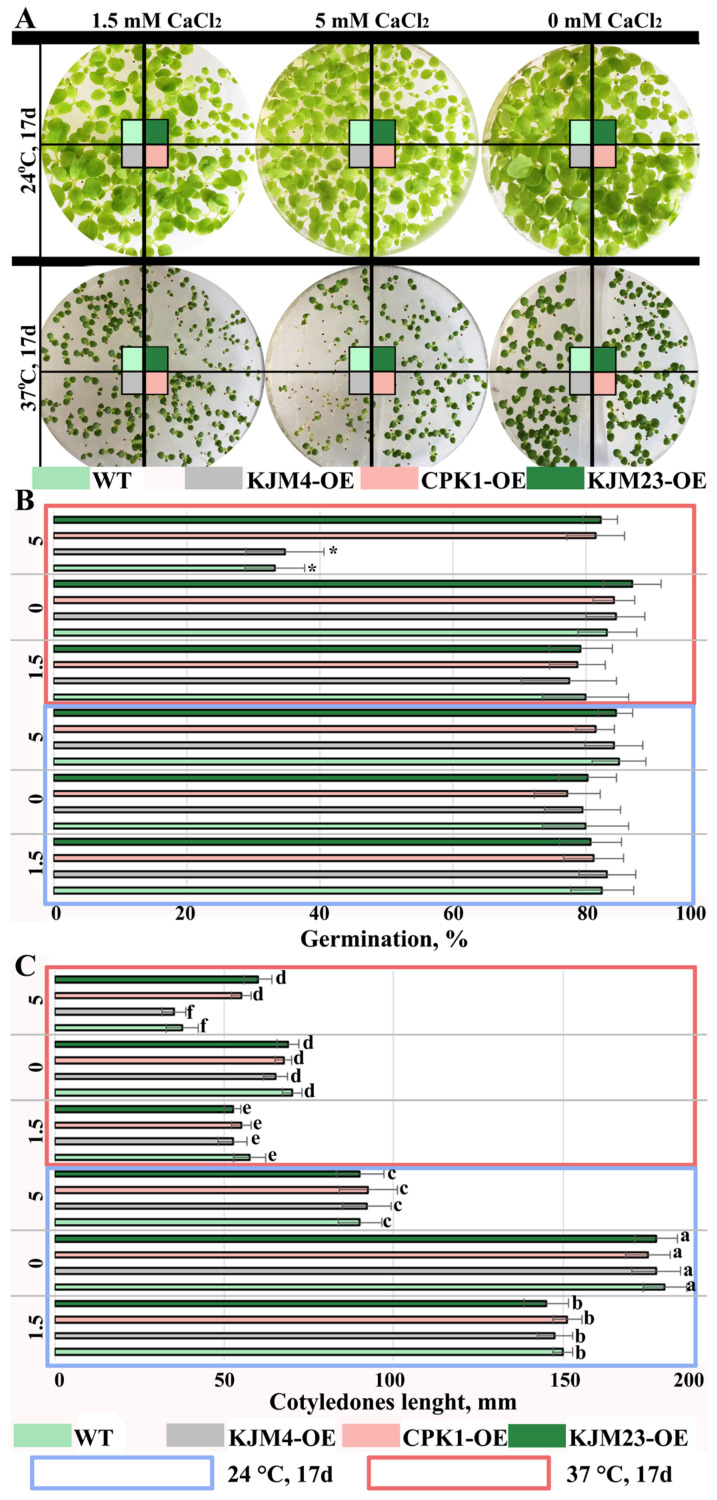
Effects of moderate heat stress on the germination of *N. tabacum* plants. Seeds of the control (WT), transformed native (CPK1-OE), mutant inactive (KJM4-OE), and constitutively active (KJM23-OE) *AtCPK1* gene plants were sterilely germinated on MS/2 agar media for 17 days. The cotyledons were photographed (**A**), and the percentage (**B**) of germination and length (mm) of the seedlings (**C**) were determined. (**A**)—Seeds germinating under control conditions (24 °C, 17 days, top); bottom: seeds germinating under stress conditions (37 °C, 17 days). For heat treatments, MS medium was supplemented with 0, 1.5, or 5 mM CaCl_2_. Three separate experiments provided the results, which are shown as the mean ± standard error of the mean. Statistically significant differences are indicated by the asterisk (**B**) or various letters (**C**) above the error bars (*p* < 0.05, Fisher’s LSD).

**Figure 5 plants-14-01032-f005:**
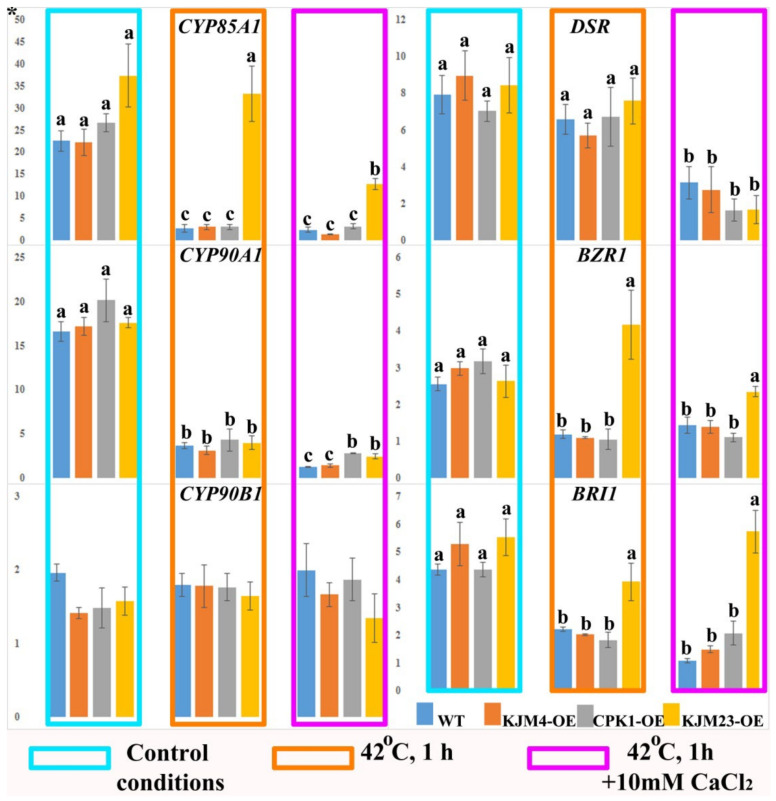
Expression of the BR biosynthesis (*NtCYPs* and *NtDSR*) and signalling (*NtBZR1* and *NtBR1*) genes in *N. tabacum* plants after heat shock treatment with or without CaCl_2_. Forty-day-old control (WT), transformed native (CPK1-OE), mutant inactive (KJM4-OE), and constitutively active (KJM23-OE) forms of the *AtCPK1* gene were subjected to temperature stress (42 °C) for 1 h with supplementation of 10 mM calcium chloride for one group. Three independent experiments (biological replicates) were used for each stress treatment, and three technical replicates were used for qPCR measurements of each biological replication. The values were computed from the qPCR data using the 2^−ΔΔCt^ technique. The asterisks above the vertical axis denote the relative expression level. Fisher’s LSD and statistically significant differences (*p* < 0.05) are indicated by the various letters above the error bars.

**Figure 6 plants-14-01032-f006:**
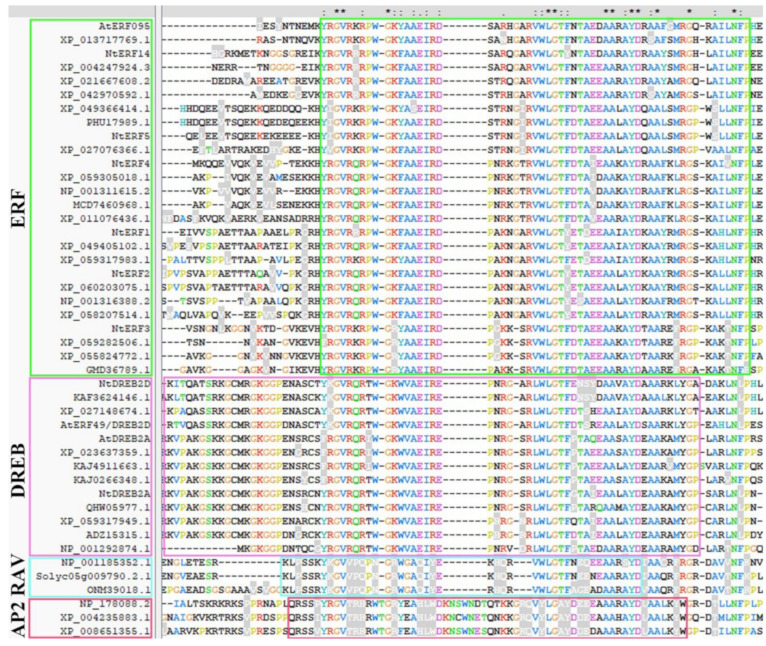
The alignment of amino acid (a.a.) sequences of AP2 family members of *N. tabacum*. NtERFs (NtERF1-5, NtERF14), homologues of *A. thaliana* DREB2A, DREB2D, ERF96, and homologues from other plants were divided into subfamilies (ERF, DREB, RAV, and AP2) and highlighted in green, pink, blue, and red blocks, respectively. The conserved single AP2 domains in ERFs and the DREB and RAV subfamilies are highlighted in green, pink, and blue blocks, respectively. The first conserved AP2 domain in the AP2 subfamily is highlighted with a red block. The AP2 family is divided into subfamilies (ERF, DREB, RAV, and AP2), and conserved domains are defined according to [31]. The amino acid (a.a.) sequences were aligned via ClusatlX, and the amino acid (a.a.) sequences with GenBank IDs are listed in Appendix A. The dashes, asterisk, dot, and colon indicate gaps, identical amino acid residues, semi-conserved and conserved substitution in all sequences used in the alignment respectively.

**Figure 7 plants-14-01032-f007:**
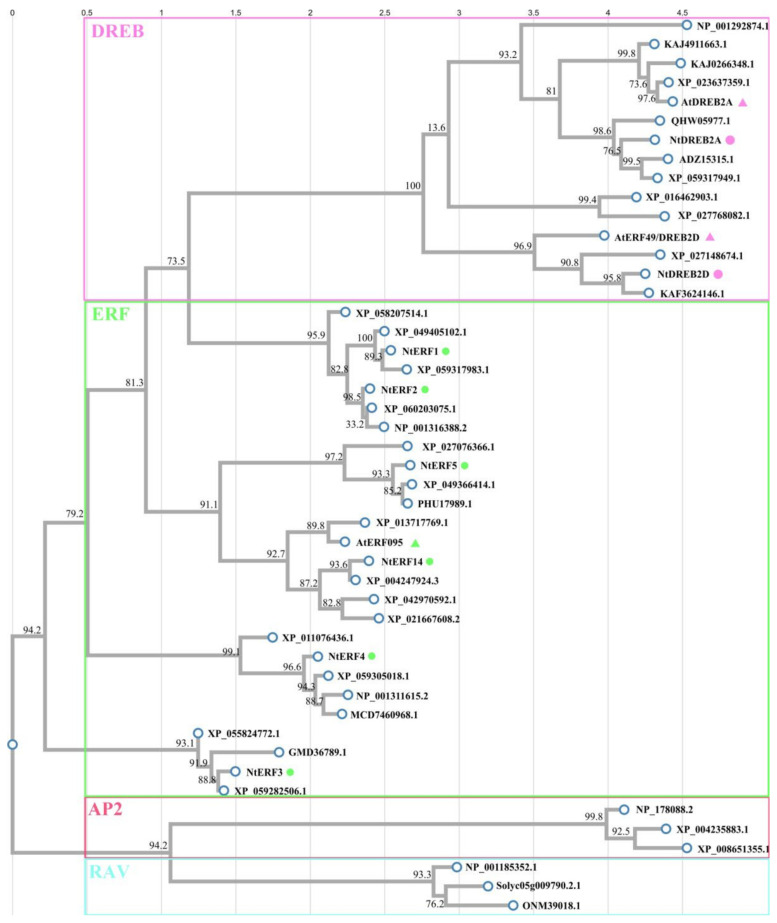
Phylogenetic relationships between AP2 family members of *N. tabacum* and those of other plants. NtERFs (NtERF1-5, NtERF14), homologues of *A. thaliana* DREB2A, DREB2D, ERF96, and homologues from other plants were divided into subfamilies (ERF, DREB, RAV, and AP2) and highlighted in green, pink, blue, and red blocks, respectively. The AP2 family members of *N. tabacum* and *A. thaliana* are designated with dots and triangles, respectively, and the colours correspond to the subfamily groups. The amino acid (a.a.) sequences were aligned via ClusatlX, the tree was constructed via the ClustalW online tool, and the amino acid (a.a.) sequences with GenBank IDs are listed in Appendix A.

**Figure 8 plants-14-01032-f008:**
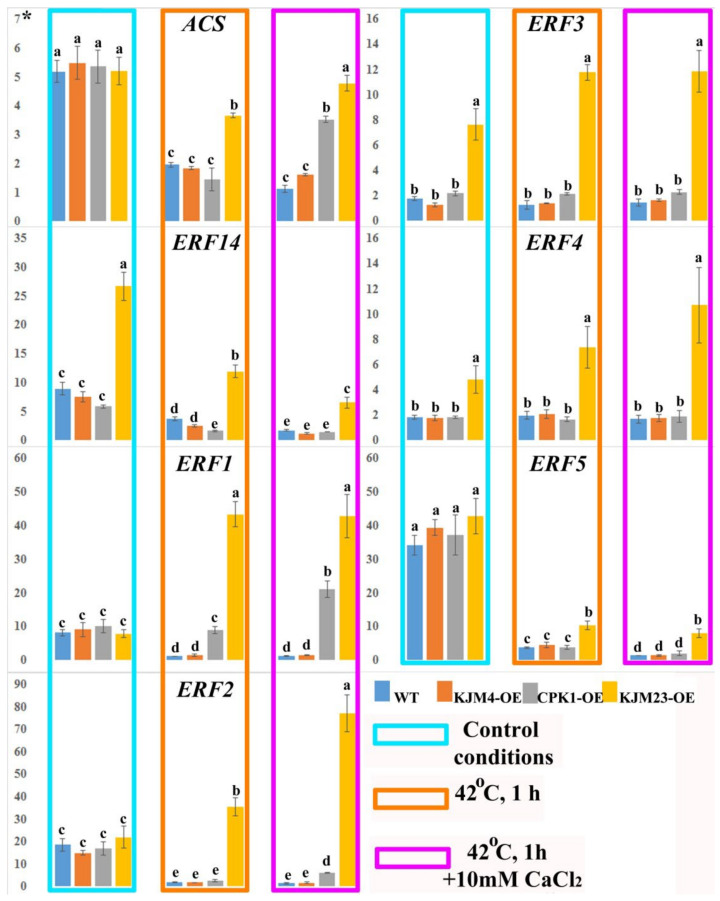
Expression of the ethylene biosynthesis (*NtACS*) and signalling (*NtERFs*) genes in *N. tabacum* plants after heat shock treatment with or without CaCl_2_. Forty-day-old, in vitro-grown control (WT), transformed native (CPK1-OE), mutant inactive (KJM4-OE), and constitutively active (KJM23-OE) forms of the *AtCPK1* gene were subjected to temperature stress (42 °C) for 1 h with supplementation of 10 mM calcium chloride for one group. Three independent experiments (biological replicates) were used for each stress treatment, and three technical replicates were used for qPCR measurements of each biological replication. The values were computed from the qPCR data using the 2^−ΔΔCt^ technique. The asterisks above the vertical axis denote the relative expression level. Fisher’s LSD and statistically significant differences (*p* < 0.05) are indicated by the various letters above the error bars.

**Figure 9 plants-14-01032-f009:**
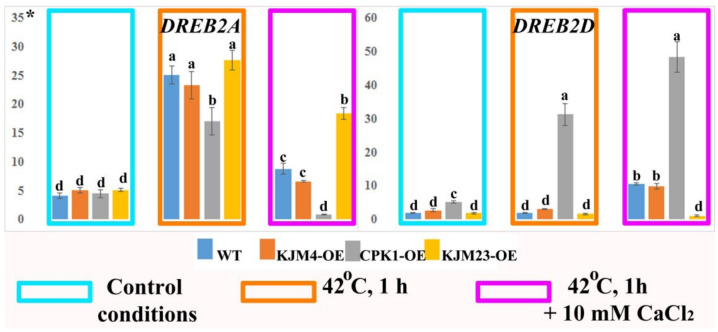
Expression of the *NtDREB* genes in *N. tabacum* plants after heat shock treatment. 40-day-old, in vitro-grown control (WT), transformed native (CPK1-OE), mutant inactive (KJM4-OE), and constitutively active (KJM23-OE) forms of the *AtCPK1* gene were subjected to temperature stress (42 °C) for 1 h with the addition of 10 mM calcium chloride for one group. Three independent experiments (biological replicates) were used for each stress treatment, and three technical replicates were used for qPCR measurements of each biological replication. The values were computed from the qPCR data using the 2^−ΔΔCt^ technique. The asterisks above the vertical axis denote the relative expression level. Fisher’s LSD and statistically significant differences (*p* < 0.05) are indicated by the various letters above the error bars.

**Figure 10 plants-14-01032-f010:**
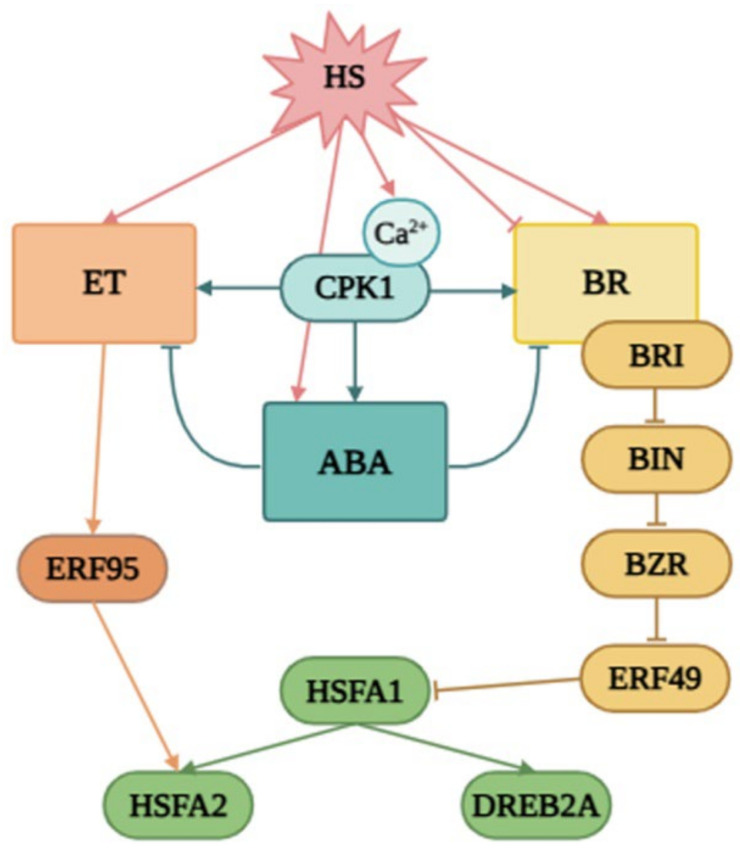
The proposed model of the role of CPK1 in the modulation of BR, ET, and ABA signalling under heat stress. Early heat exposure causes an increase in intracellular calcium concentration [45], which sets off CDPK-regulated cascades. BR may be inhibited under heat stress, leading to the accumulation and activation of BIN2, which leads to the activation of *ERF49* (DREB2D) expression [9]. In another work, the authors proposed that BR is induced by heat stress, leading to the activation of BZR1, which represses *ERF49* expression. *Arabidopsis* HSFA1s function as “master regulators” and are induced under heat stress to directly regulate the expression of downstream *HEAT STRESS RESPONSIVE* (*HSR*) genes or indirectly by activating the AP2/ERF TF *DREB2A* [45]. The expression of *ERF95* or *ERF97* enhances basal thermotolerance in *Arabidopsis* [11]. This model was described in detail by [15]. As we previously reported, overexpression of the native form of the AtCPK1 gene led to a heat-induced increase in the expression of genes involved in ABA biosynthesis, in contrast with the calcium-independent form KJM23.

## Data Availability

The original contributions presented in this study are included in the article/Appendix A. Further inquiries can be directed to the corresponding author.

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
