# Peer review of "Differential Modulation of Brassinosteroid and Ethylene Signalling Systems by Native and Constitutively Active Forms of the AtCPK1 Gene in Transgenic Tobacco Plants Under Heat Stress"

_plants, 2025, doi:10.3390/plants14071032_

Round 1
Reviewer 1 Report
Comments and Suggestions for Authors
The manuscript title “Differential modulation of brassinosteroid and ethylene signalling systems by native and constitutively active forms of the AtCPK1 gene in transgenic tobacco plants under heat stress” is conducted well and have scientific worth. This research provides insights into the molecular processes associated with heat stress in plants and highlights the potential of intradomain modifications of CDPKs as tools for both functional studies and bioengineering applications. Understanding these interactions may lead to improved strategies for enhancing plant resilience to heat stress.
Comments for authors are as follows:
- Line 68: “Many transcription factors (TFs) mediate” and Line 70: “Plant-specific transcription factors (TFs) of the” if the authors mention the abbreviation first time, then don’t need to mention again and again. Please check all the abbreviations and make consistency.
- Line 140: “grown in vitro [28].” Avoid using references in the results section. Lines 181, 182.
- Line 159: “nonfluorescent chlorophyll catabolites (NCCs) were detected (Veremeichik et al., 2025).” This is “(Veremeichik et al., 2025)” inappropriate use to references, please use cite the references according to the Journal rules.
- Line 202: “significant differences (P < 0.05, Fisher’s LSD).” Have you performed ANOVA? If yes, then include the ANOVA table.
- Line 203: “(Figure 3, B)” the figure citation in the text should be (Figure 3B), modify all the figure citations… Line 222 figure citation should be (Figure 4A, B). Line 533: “CaCl2.” 2 should be superscript.
- Line 575: “The oligonucleotide pairs used as housekeeping genes and for the analysis of” which housekeeping genes was used? Mention here….
- Line 588: “ANOVA and multiple” where is ANOVA table?
- Section 4.6: Which software was used to make graphs is not mentioned.
Author Response
Dear Reviewer #1 we express our sincere gratitude for the positive feedback about our work. Your comments are very important for the manuscript.
The manuscript title “Differential modulation of brassinosteroid and ethylene signalling systems by native and constitutively active forms of the AtCPK1 gene in transgenic tobacco plants under heat stress” is conducted well and have scientific worth. This research provides insights into the molecular processes associated with heat stress in plants and highlights the potential of intradomain modifications of CDPKs as tools for both functional studies and bioengineering applications. Understanding these interactions may lead to improved strategies for enhancing plant resilience to heat stress.
Comments for authors are as follows:
- Line 68: “Many transcription factors (TFs) mediate” and Line 70: “Plant-specific transcription factors (TFs) of the” if the authors mention the abbreviation first time, then don’t need to mention again and again. Please check all the abbreviations and make consistency.
- Answer: Corrected
- Line 140: “grown in vitro [28].” Avoid using references in the results section. Lines 181, 182.
- Answer: Corrected, transferred in the Introduction section
- Line 159: “nonfluorescent chlorophyll catabolites (NCCs) were detected (Veremeichik et al., 2025).” This is “(Veremeichik et al., 2025)” inappropriate use to references, please use cite the references according to the Journal rules.
- Answer: Corrected
- Line 202: “significant differences (P < 0.05, Fisher’s LSD).” Have you performed ANOVA? If yes, then include the ANOVA table.
- Answer: Included in Supplementary material
- Line 203: “(Figure 3, B)” the figure citation in the text should be (Figure 3B), modify all the figure citations… Line 222 figure citation should be (Figure 4A, B).
- Answer: Corrected
- Line 533: “CaCl2.” 2 should be superscript.
- Answer: Corrected
- Line 575: “The oligonucleotide pairs used as housekeeping genes and for the analysis of” which housekeeping genes was used? Mention here….
- Answer: Corrected
- Line 588: “ANOVA and multiple” where is ANOVA table?
- Answer: Included in Supplementary material
- Section 4.6: Which software was used to make graphs is not mentioned.
- Answer: Corrected
Reviewer 2 Report
Comments and Suggestions for Authors
attachment

Author Response
Dear Reviewer #2 we express our sincere gratitude for the positive feedback about our work. Your comments are very important for the manuscript.
Figures 1, 3, 4 have been corrected
Line 495: HsFA1s ® HSFA1s
Corrected
Line 533: 10 mM CaCl2 ® 10 mM CaCl2
Corrected
Line 542: 5 mM CaCl2. ® 5 mM CaCl2.
Corrected